# Six Steps towards a Spatial Design for Large-Scale Pollinator Surveillance Monitoring

**DOI:** 10.3390/insects15040229

**Published:** 2024-03-26

**Authors:** Niels Hellwig, Frank M. J. Sommerlandt, Swantje Grabener, Lara Lindermann, Wiebke Sickel, Lasse Krüger, Petra Dieker

**Affiliations:** 1Thünen Institute of Biodiversity, Bundesallee 65, 38116 Braunschweig, Germany; frank.sommerlandt@thuenen.de (F.M.J.S.); swantje.grabener@thuenen.de (S.G.); lara.lindermann@thuenen.de (L.L.); wiebke.sickel@thuenen.de (W.S.); lasse.krueger@thuenen.de (L.K.); petra.dieker@thuenen.de (P.D.); 2National Monitoring Centre for Biodiversity, Federal Agency for Nature Conservation, Alte Messe 6, 04103 Leipzig, Germany

**Keywords:** biodiversity monitoring, sampling design, sampling method framework, agroecosystems, wild bees, spatial sampling

## Abstract

**Simple Summary:**

Insect pollinators are vital players in agroecosystems, but representative data on long-term trends are scarce. Large-scale monitoring programmes are needed to fill the data gaps on long-term pollinator trends in agroecosystems. Here, we ask how spatial sampling for scientifically robust pollinator monitoring should be designed, meaning which aspects in particular need to be considered. We present six steps to establish a sample set suitable for such a monitoring programme, together with specific plots for pollinator surveys within landscape-scale sampling units. We focus on the sampling design of the recently developed “Wild bee monitoring in agricultural landscapes of Germany” programme. Our six-step framework can serve as a model for the spatial sampling design of future large-scale pollinator monitoring programmes.

**Abstract:**

Despite the importance of pollinators to ecosystem functioning and human food production, comprehensive pollinator monitoring data are still lacking across most regions of the world. Policy-makers have recently prioritised the development of large-scale monitoring programmes for pollinators to better understand how populations respond to land use, environmental change and restoration measures in the long term. Designing such a monitoring programme is challenging, partly because it requires both ecological knowledge and advanced knowledge in sampling design. This study aims to develop a conceptual framework to facilitate the spatial sampling design of large-scale surveillance monitoring. The system is designed to detect changes in pollinator species abundances and richness, focusing on temperate agroecosystems. The sampling design needs to be scientifically robust to address questions of agri-environmental policy at the scales of interest. To this end, we followed a six-step procedure as follows: (1) defining the spatial sampling units, (2) defining and delimiting the monitoring area, (3) deciding on the general sampling strategy, (4) determining the sample size, (5) specifying the sampling units per sampling interval, and (6) specifying the pollinator survey plots within each sampling unit. As a case study, we apply this framework to the “Wild bee monitoring in agricultural landscapes of Germany” programme. We suggest this six-step procedure as a conceptual guideline for the spatial sampling design of future large-scale pollinator monitoring initiatives.

## 1. Introduction

Wild pollinators are a target group in conservation programmes across agricultural landscapes [1,2]. Due to their relevance not only for biological diversity, but also for agroecosystems and food production, their abundance and richness are often regarded as important farmland biodiversity indicators. They are thus be included in proposed monitoring schemes [3,4,5,6]. However, large-scale standardised pollinator monitoring programmes are still rare (but see [7,8,9,10]). Continuous declines in pollinators and other flying insects across Europe (e.g., [11,12]) have recently led government authorities to commission concepts for monitoring programmes [10,13,14,15]. Many researchers are now challenged to design monitoring programmes to reliably report future trends in species abundance and richness, and to relate the resulting data to ecological and anthropogenic pressures [16,17]. A reverse in pollinator declines with targeted restoration and conservation measures can only be achieved based on a detailed understanding of ecological processes and anthropogenic effects on pollinators at the landscape level. Hence, a landscape-level approach to pollinator monitoring is necessary, i.e., based on sampling units that allow a combined analysis of pollinator data and environmental data. Persistent trends in population sizes should be distinguishable from short-term fluctuations [18], for example due to seasonal weather conditions affecting population development.

Several published guidelines explain the development of environmental monitoring programmes and aspects to be considered in sampling designs. For example, de Gruijter et al. [19] presented an overview of natural resource monitoring covering multiple sampling strategies and a global guideline for the design process. Lindenmayer et al. [20] compiled a checklist on important considerations for monitoring programmes including the design, and Reynolds et al. [21] provided a road map for designing and implementing biological monitoring programmes, with the sampling design covered as one of ten successive steps. However, for non-statisticians in particular, further explanation is needed to develop a scientifically robust spatial sampling design. A separate, specific road map for a systematic spatial sampling design is needed, especially in light of the increasing interest in pollinator monitoring initiatives (e.g., [10]). We focus on surveillance monitoring programmes with long-term data on pollinator abundance and richness that can be linked to environmental data to derive drivers and assess impacts of pollinator trends.

Several questions need to be addressed in the sampling design [22], i.e., where, when and how often to sample; what and how to sample; and who should carry out the sampling and at what cost. The sample set must be representative of the area to be monitored, and biases in the sample set should be minimised or accounted for in the analysis.

Since many requirements represent decisions that need to be made prior to the spatial sampling design (Figure 1), they are out of the scope of this article and are only briefly described here. First, the objectives of the pollinator surveillance monitoring need to be specified, for example, the pollinator taxa and other attributes to be investigated [21].

Second, the metrics and indicators to be reported should be defined, ideally integrating political needs and a cost–benefit analysis (e.g., [23]).

Third, specific survey methods are recommended to monitor the selected groups of pollinators (e.g., [24,25,26,27,28,29]). Those methods may differ in taxonomic resolution, plot sizes, survey duration and frequency, seasonal coverage and species detectability.

Fourth, pollinator assemblages and their habitats are closely linked to the landscape context at various spatiotemporal scales [30,31,32,33]. This is mainly associated with the availability of suitable floral resources and further taxon-specific requirements such as nesting resources [34]. Thus, the sample set needs to cover the landscape heterogeneity as well as climatic variability across the monitoring area.

Fifth, the intended statistical analysis of pollinator trends requires preliminary considerations of the statistical power for trend detection as a basis for determining the sample size. Moreover, to define the spatial and temporal scales of the sampling design, the planned level of harmonisation of the pollinator and environmental data needs to be clarified in advance.

Sixth, the capacities for field surveys and laboratory analyses available and which survey plots are accessible and feasible must be assessed. A strategy for data management and storage should be conceived (e.g., a database management system) to allow efficient workflows and prevent data loss.

Once these requirements have been clarified, the next challenge is the spatial sampling design itself. Inspired by previous studies [19,20,21], we present a six-step approach to provide taxonomists and conservationists with a guideline for the development of a representative spatial sample set corresponding to their own set of requirements (according to Figure 1). Specifically, this guideline addresses the following questions, faced by scientists in developing monitoring schemes:How can spatial sampling units be defined to be suitable for monitoring pollinator assemblages at the landscape level, i.e., the scale at which environmental factors affect pollinators?;How many sampling units are needed to achieve sufficient statistical power for the analysis of pollinator trends based on assumptions about species abundance and richness?;Where should those sampling units be placed to represent the gradient of landscapes across the monitoring area?;What are suitable temporal intervals between sampling periods to meet the monitoring requirements?

As an example of the application of the six-step procedure, we present a case study on wild bee surveillance monitoring in the agricultural landscapes of Germany, which has been developed and tested for implementation since 2019. Based on the six-step procedure and the case study, we discuss (1) aspects to be considered for learning and revising the sampling design and (2) perspectives to gain knowledge on the spatiotemporal trends of pollinators across scales.

## 2. Conceptual Framework for the Spatial Sampling Design

Following the terminology in spatial sampling introduced in Table 1, a six-step procedure is applied to the above challenges in the sampling design (Figure 2). Spatial sampling units within the monitoring area are defined in Steps 1 and 2. Then, Steps 3, 4 and 5 draw a subset out of those sampling units and determine the temporal sampling intervals. In Step 6, the locations of the pollinator surveys are selected within each sampling unit. This six-step procedure for the spatial sampling design should be considered as part of the overall development of the monitoring programme, including all steps according to Reynolds et al. [21] and the clarification of the monitoring requirements (Figure 1). Hence, it may be iteratively revisited during the design, implementation and revision of pollinator monitoring programmes (Appendix A).

### 2.1. Step 1: Define Spatial Sampling Units Based on Monitoring Requirements

A main task of any landscape-level pollinator monitoring approach is to detect the spatial patterns and temporal trends of pollinators and how they respond to landscape characteristics, dynamics and management. This means that pollinator data should be acquired on the landscape level, where they can be related to data on landscape heterogeneity and environmental changes. The spatial scale of the sampling units affects the monitoring results in terms of biodiversity metrics and environmental drivers [33,35]. However, until now, there has been no common practice in the size of the sampling units used in pollinator monitoring. Landscape quadrats of 1 km × 1 km have often been used in pollinator monitoring approaches (e.g., [7,36,37]) and further studies on landscape-level pollinator assemblages (e.g., [38,39]). Recently, 3 km × 3 km quadrats have been used for landscape-level assessments of wild bees in the context of biodiversity measures in agricultural landscapes of Germany [40] and for monitoring the landscape-level impacts of agri-environment schemes on wild bees and other taxa in England [41]. Generally, it would be helpful to harmonise the sampling units used for different pollinator monitoring programmes. Across Europe, the regular grid of the Land Use and Coverage Area frame Survey (LUCAS grid) is a useful basis for the definition of landscape quadrats as sampling units. The current LUCAS grid consists of 1,033,759 sample points, covering all member states of the European Union in a grid width of 2 km [42]. Apart from periodic surveys on land use and coverage, the LUCAS grid has become increasingly popular as the spatial basis for monitoring schemes, for example, on pollinators (EU-PoMS, [10]) and on biodiversity in agricultural landscapes (EMBAL, [43]).

### 2.2. Step 2: Define and Delimit the Monitoring Area

The area for pollinator monitoring needs to be defined according to the predefined monitoring objectives and political context. For example, this could be the total area of a specific country or physiographic region, or agricultural landscapes within a specific region. Furthermore, considerations of survey methods, environmental context and practical aspects may require a specific preselection of the sampling units from Step 1. It is crucial to define clear preselection criteria for the location and landscape composition of the sampling units. Based on such criteria, the result of Step 2 is a subset of the sampling units determined in Step 1 (i.e., all sampling units that fulfil the criteria), which is then further considered in Steps 3–6.

### 2.3. Step 3: Decide on the General Spatial Sampling Strategy

After the sampling units have been defined in Steps 1 and 2, a general strategy is needed to select a sample set. In general, spatial sampling follows the design-based or the model-based approach [19,44]. The design-based approach is realised with random sampling, which allows for valid estimates of pollinator population parameters such as the number of wild pollinator species and individuals within the monitoring area. The most common techniques are simple random sampling, stratified random sampling and systematic random sampling [19]. Stratified random sampling appears useful in the case of large environmental gradients across the monitoring area (e.g., due to topography) or in the case of specific political needs, where landscapes or administrative regions of different strata should be sampled with predefined effort (e.g., [45]). Caution should be exercised when defining strata according to agricultural land use (e.g., [46]) or when spatially prioritising specific land use classes (e.g., [9]) because land use and management are likely to vary over time. Systematic random sampling ensures a spatially even distribution of sampling sites across the monitoring area. In contrast to the design-based approach, the model-based approach relies on a statistical model that explains the distribution of the population parameters such as diversity or abundance based on environmental data. In the case of a valid model, the spatial distribution of the sampling sites can be optimised, integrating the information about the distributions of the environmental covariates, for example with Conditioned Latin Hypercube Sampling or Feature Space Coverage Sampling [47,48,49,50].

With regard to pollinators, numerous studies have analysed and modelled the effects of environmental influences (e.g., [51,52,53,54]). However, the validity of the reported relationships between pollinators and their environment has not been tested for large areas over the long term. Therefore, to avoid any site-selection bias [55,56], the use of a random sampling strategy appears advisable for long-term pollinator monitoring.

### 2.4. Step 4: Determine the Sample Size

The sample size is a main factor in obtaining meaningful results from a biodiversity monitoring programme. It depends on the spatial variability of the population of the monitored taxa [22,39,57]. Pollinator population parameters derived from the sample (i.e., mean and variance of species abundance and richness) should generally have low estimate errors. This means that in the case of a surveillance monitoring programme it is essential to achieve a high probability (or high power of the related hypothesis test) that certain temporal trends in the population parameters can be detected with high statistical significance (e.g., [23,58,59]). For example, the sample size could be determined so that it would be likely to detect a 5% or 10% decline in species richness between two sampling events (target statistical power is often ≥80%). Using assumptions about the expected values and variances of species richness, the required sample size can be calculated by computer simulations [60,61].

### 2.5. Step 5: Specify Spatial Sampling Units per Sampling Interval

From the sampling units defined in Step 1 and preselected in Step 2, a subset of the planned pollinator monitoring, that is, the sample set, can be identified using the sampling strategy selected in Step 3 and the sample size calculated in Step 4. Hence, the result of Step 5 is the spatial arrangement of the sampling units that are intended for the monitoring surveys. Sampling intervals could, for example, be defined as one year to optimally cover interannual variation [62]. Depending on the survey method, every sampling interval potentially comprises several complementary surveys throughout the season. In the case of an alternating sampling approach (mainly for reasons of cost and effort), different subsets of the sample set can be selected to be monitored per sampling interval.

### 2.6. Step 6: Specify Survey Plots within Each Spatial Sampling Unit

In a landscape-level approach to pollinator monitoring, all the sampling units belonging to the sample set specified in Step 5 usually represent landscape quadrats or comparable sampling units. The exact locations of pollinator surveys need to be specified according to the survey methods, so that acquired pollinator data are representative of the whole landscape. For example, to receive a good representation of the present pollinators, a single transect with its linear shape should be placed crossing as many habitat types as possible according to their presence in the sampling unit. As another example, multiple point locations are needed in the case of any kind of trap. Their number per sampling unit should be high enough to represent the pollinator community at the landscape level, i.e., the scale of the sampling units. Across every sampling unit, traps should be located either randomly or in a model-based approach based on landscape features, according to the predefined monitoring objectives. Depending on the target pollinators, a minimum distance between survey plots needs to be kept in order for them to count as independent.

## 3. Case study: Monitoring Cavity-Nesting Wild Bees in Agricultural Landscapes in Germany

A nationwide long-term monitoring programme of wild bees in agricultural landscapes has been designed and tested since 2019 (https://wildbienen.thuenen.de/, accessed on 25 January 2024; https://www.agrarmonitoring-monvia.de/en/, accessed on 25 January 2024). This case study focuses on one surveillance monitoring module where artificial nesting aids are used to detect above-ground cavity-nesting wild bees. This module was conceived in 2019–2021 and implemented for test purposes in 2022–2023 (with data collection ongoing in 2024). The survey method of this module is standardised nesting aids, i.e., a kind of trap nest designed for the non-lethal, observer-independent detection of wild bees. Data are collected by volunteers and validated by experts [63]. Wild bee taxa are identified from photos of their brood cells, and the residues in the nesting aids are sampled for environmental DNA (eDNA) analysis after the bees have hatched. These eDNA analyses will provide data for a more precise identification of the colonising taxa at the species level and additional data on the food and nesting resources used by wild bees [64].

A nationwide wild bee monitoring programme cannot be established without considering the country-specific demands and requirements. In Germany, biodiversity monitoring is a task assigned to the federal states. Thus, a nationwide approach should also be scalable according to the needs and activities of federal states or even at more regional levels. Furthermore, to get the most out of a nationwide wild bee monitoring programme, synergies with international monitoring activities should be addressed, for example with the European Pollinator Monitoring Scheme (EU-PoMS, [10]) and the European Monitoring of Biodiversity in Agricultural Landscapes (EMBAL, [43]).

In the following, we describe in detail how we devised the spatial sampling design of the case study on cavity-nesting wild bees in the agricultural landscapes of Germany, following the six-step procedure described above.

### 3.1. Step 1: Define Spatial Sampling Units Based on Monitoring Requirements

As the landscape affects wild bees on multiple spatial scales, especially up to 3 km [33], we defined the sampling units as 3 km × 3 km landscape quadrats. This is in line with recent landscape-level assessments of wild bees in the agricultural landscapes of Germany (e.g., FInAL project, https://www.final-projekt.de/en/landscape-laboratories, accessed on 25 January 2024; [40]), These landscape quadrats were oriented according to the pan-European LUCAS grid [42]. The sum of all the LUCAS grid points in Germany is 89,341 points, and each of these points was considered the central point of a potential 3 km × 3 km landscape quadrat (Figure 3).

### 3.2. Step 2: Define and Delimit the Monitoring Area

Agricultural areas cover around 50% (data from 2020) of the total area of Germany [65]; thus, large parts of Germany belong to the agricultural landscape. The target area for the wild bee monitoring was delimited considering the agricultural areas (object type “AX_Landwirtschaft”, [66]) as classified by the Basic Digital Landscape Model (DLM) [67]. Each of the landscape quadrats was characterised as an agricultural or non-agricultural landscape according to a threshold of 30% cover of agricultural area in the 3 km × 3 km quadrat. Quadrats characterised as a non-agricultural landscape were often dominated by other land use types such as forests and urban areas. Moreover, in quadrats with less than 30% cover of agricultural areas it appeared difficult or impossible to find enough suitable, independent locations for nesting aids.

By preselecting the agricultural landscape quadrats, the population of potential sampling units was reduced from 89,341 to 60,512, corresponding with 67.7% of the total number of LUCAS grid points across Germany. Most of the excluded grid points were located in the large forest areas of the uplands, the Alps and large urban agglomerations (Appendix A).

### 3.3. Step 3: Decide on the General Spatial Sampling Strategy

Environmental effects on cavity-nesting wild bees, which could serve as factors in a model-based spatial sampling approach, have been described by multiple studies across Central Europe (e.g., [68,69,70]). However, due to limited spatiotemporal scales, the findings from those studies are not generalisable for the whole of Germany over the long term. Thus, the spatial sampling design of the wild bee monitoring programme relied on a random sampling strategy.

As the diversity of the agricultural landscapes across Germany should be covered by the sample set and as volunteers from all over Germany should have similarly easy access to the sampling units, the landscape quadrats were drawn as a systematic random sample based on a standardised grid width (i.e., equal distances between the LUCAS grid points). This means that any of the preselected landscape quadrats was selected randomly and the other quadrats of the sample were added automatically as multiples of equal distance in each of the directions north, east, south and west.

### 3.4. Step 4: Determine the Sample Size

Power analyses to determine the sample size for the wild bee monitoring followed the approach by Johnson et al. [60], using the statistical software R, version 4.0.2 [71] and the function *sim.glmm* [60]. We considered the species richness of cavity-nesting wild bees per landscape quadrat and year as response variable. Assumptions about the species richness were derived from previous field studies across Germany and nearby places (Appendix A). Based on 15 species of cavity-nesting wild bees at the expected value per landscape quadrat, the simulation results showed a consistent relationship between the number of sampled landscape quadrats and the statistical power to detect a 3% decline in cavity-nesting wild bee species at *p* < 0.05, independent of the variance (Figure 4). In total, a sample size of about 1000 landscape quadrats (i.e., sampling units) was necessary to achieve a power of 80%.

### 3.5. Step 5: Specify Spatial Sampling Units per Sampling Interval

The target sample set specified based on Steps 1–4 comprised 950 landscape quadrats across Germany, arranged in a grid with 16 km width between the LUCAS grid points at the centre of the quadrats (Figure 5a). Gaps within the grid resulted from the preselection of landscape quadrats in the agricultural landscape (Step 2). A full list of all the landscape quadrats selected for the wild bee monitoring programme is provided in Appendix A.

In general, the wild bee monitoring programme has been designed to analyse trends. However, to avoid biased abundance data caused by philopatric behaviour that is known among many wild bee species (e.g., [72]), or even the artificial establishment of wild bee populations and locally increased pressure by parasitoids, the sampling design for monitoring cavity-nesting wild bees uses an alternating approach. This means that nesting aids are installed only every second year per landscape quadrat. Moreover, considering a cost-effective analysis of colonising wild bees and nest-building residues via eDNA [64], molecular analyses are scheduled only every fourth year per landscape quadrat.

In case political needs require more extensive sampling of specific areas to obtain regional pollinator trend estimates with high confidence, the grid width can be reduced to obtain a higher density of sampling units. For example, by extension based on a grid width of 8 km, the sample size for the federal state of Saxony-Anhalt can be increased from 75 to 261 (Figure 5b).

### 3.6. Step 6: Specify Survey Plots within Each Spatial Sampling Unit

Considering the different flight distances of cavity-nesting wild bees [73,74] and based on previous landscape-level studies [63], we chose six locations for nesting aids per landscape quadrat (Figure 3). At these locations, two nesting aids are placed at different time points, one at the end of March and one at the end of May, so that wild bee species with different phenologies are provided with sufficient nesting opportunities early and late in the season. This design ensured that the assemblages of cavity-nesting wild bees were well represented throughout the landscape quadrat on the one hand and that the six locations of the nesting aids were independent from each other on the other hand (Lindermann, unpublished data). The exact locations of the nesting aids (blue circles in Figure 3) were determined within the landscape quadrats in a semi-automated procedure. For every landscape quadrat in the sample set, the initial locations for the nesting aids were determined with the following algorithm:Extract boundaries of areas under agricultural land use per 3 km × 3 km landscape quadrat (Basic DLM, [67]).Randomly sample six sites across all boundaries from Step (A) with a minimum distance of 250 m to the margin of the landscape quadrat.Calculate all pairwise distances between the six sites for nesting aids.Repeat Steps (B) and (C) 10,000 times.Select that option (of all 10,000 options) that maximises the smallest of all distances calculated in Step (C).

Step (E) ensures that all the survey plots are arranged with distances >500 m such that the cavity-nesting wild bees are surveyed independently. The algorithm as implemented in R, version 4.0.2 [71], can be found in the Appendix A (see also details in the Appendix A). The final locations of the nesting aids were then adjusted based on orthophotos (DOP20, BKG; Google Earth) and the volunteers’ local field knowledge.

## 4. Discussion

### 4.1. Learning and Revising the Sampling Design

A successful long-term monitoring programme relies on continuous learning about the target pollinator groups. This learning process implies reformulating conceptual models, which are the basis of understanding the monitored system, and revising and reimplementing the sampling design accordingly [17,21,75]. For example, Henriques et al. [76] have shown how to evaluate sampling suggestions for surveillance monitoring of the Red List Index based on new data after twelve years. In a long-term monitoring programme of plant species in the Netherlands, van Strien et al. [77] reconsidered their monitoring concept after an eight-year survey round to improve the citizen science-based approach. Based on 20 years of experience from the Norwegian Monitoring Programme for Agricultural Landscapes, Stokstad & Fjellstad [78] described how the sampling method was adapted to comply with new political demands. Thus, the overall set of 950 landscape quadrats presented for the wild bee monitoring programme in the agricultural landscapes in Germany (Figure 5a) should be considered as the initial sample, to be revised based on future findings or changing requirements.

The final locations of the sampling units and survey plots specified in Steps 5 and 6 of the case study have been the basis for test surveys with volunteers in the pilot phase (2022–2023, Phase II in Appendix A). The regular distribution of landscape quadrats allows volunteers from all over Germany to access sampling units nearby. By integrating the volunteers’ local knowledge and their experience of the exploration of the landscape quadrats, the exact locations of the nesting aids can be adjusted from the algorithm-based suggestions (Step 6). The volunteers’ feedback can also be valuable to further develop the algorithm to optimise initial suggestions in the future, possibly also integrating further geodata on roads and field paths. From our first experiences of the pilot phase, the spatial sampling design has proved to be suitable for application in the case study. If maintained in the long term, the data acquired from the nesting aids in the landscape quadrats are a valuable basis for predicting cavity-nesting wild bee trends.

The assumptions about the expected values and variances of species richness for power analyses were based on sparse data (Step 4 in the case study), so that possible spatial variations in trend estimates could not be included, although these might be important [79]. Therefore, an extension of the sample might be desirable based on future monitoring data. Systematic random sampling offers a flexible framework for any extension of the sample (e.g., grid widths of 8 km or 4 km). This might be also interesting to address new research questions for specific parts of the monitoring area in the future. Additionally, a systematic subsample (e.g., grid width of 32 km, 48 km or 64 km) could become relevant for further monitoring modules with a higher sampling effort, for example, for the survey of ground-nesting wild bee species. Although the presented case study treats only a small part of the pollinator fauna, decisions in the spatial sampling design are comparable to other pollinator taxa, considering their specific population metrics and habitat requirements.

### 4.2. Implications for Large-Scale Pollinator Monitoring

Pollinators include a large diversity of insect taxa. This diversity is reflected by a large variety of habitat requirements in terms of nesting or oviposition sites, food preferences and mating sites [80,81,82]. Different pollinator groups have been identified as suitable indicator taxa for large-scale monitoring schemes, for example, solitary bees, bumblebees, butterflies and hoverflies [83,84,85]. Our framework can be applied to all those taxa, with different decisions in the spatial sampling design process. For example, the size of the sampling units determines the spatial scale at which landscape effects can be evaluated. Therefore, sampling units should be large enough to cover the foraging range of species of interest. Landscape effects on pollinators have nevertheless been demonstrated at various spatial scales independent of their foraging range [33,86,87].

In general, different monitoring questions, political needs and targets, and environmental settings require different monitoring approaches and also different spatial sampling techniques [19,21]. For example, to answer specific questions with targeted monitoring, such as the effectiveness of agri-environmental measures on biodiversity, a stratified or model-based sampling approach is often suitable [38,88,89]. For long-term surveillance monitoring programmes, we recommend use of the design-based approach, applying a random sampling strategy. The acquisition of unbiased data allows for additional analyses of future research questions, possibly using a refined subsample. Furthermore, following the design-based approach on an international grid, such as the LUCAS grid, enables connections to further monitoring programmes. Such synergies could be helpful to address overarching research questions in the sense of integrative and extensive biodiversity monitoring [90,91,92].

Large-scale surveillance monitoring approaches such as that used in the case study require that the sample is spatially unbiased across the monitoring area (or that information on biases is available). Those approaches have been partly criticised because of the lack of questions or hypotheses addressed by the design [93]. However, the presented spatial sampling design of the case study on wild bee monitoring can be refined for question-based monitoring approaches for pollination performance and agri-environmental measures. Moreover, the random sample allows for data analyses with regard to research questions and hypotheses that may emerge from future developments or possibly even from the data of the surveillance monitoring themselves [93].

The need for large-scale pollinator monitoring programmes has been increasingly stressed during the last years [23,59,94]. Recently, the European Pollinator Monitoring Scheme (EU-PoMS) has been proposed [10], now providing a framework for the development of national pollinator monitoring programmes. To achieve an international, harmonised knowledge base, national approaches such as the wild bee monitoring programme in the agricultural landscapes of Germany, as presented in the case study, should be spatially designed according to the pan-European LUCAS grid as suggested by EU-PoMS [10]. Together with the results of the ongoing EU project SPRING (Strengthening pollinator recovery through indicators and monitoring, https://wikis.ec.europa.eu/display/EUPKH/SPRING+project, accessed on 25 January 2024), the presented framework could be used as the basis for a prospectively coherent continent-wide pollinator monitoring programme. Additionally, the landscape-based approach to pollinator monitoring has high potential to integrate the monitoring of pollinator organisms with the monitoring of related habitats. For example, field data on pollinators could be connected to the growing amount of easily available data from remote sensing and other earth observation data [95,96,97]. This will be important to better understand the environmental drivers of biodiversity changes at the landscape scale and to enhance landscape management [98].

In total, we suggest the presented six-step procedure as a guideline for the spatial sampling design of large-scale pollinator monitoring programmes. When implemented over the long term, concepts such as the wild bee monitoring programme in the agricultural landscapes of Germany and further pollinator monitoring activities will provide a scientific basis to advise policy-makers by reporting on long-term pollinator trends and associating these with changes in landscapes and climate. Furthermore, monitoring data can lead to a better understanding of the causes of pollinator declines and their relationship to land use change or environmental change. As a perspective, such monitoring programmes can also contribute valuable data to indicators of farmland biodiversity and pollination services in the European and global contexts [99,100].

## 5. Conclusions

This study provides guidance on the spatial framework for large-scale, landscape-level, long-term pollinator surveillance monitoring programmes. The conceptual guideline includes six steps for a spatial sampling design considering the requirements of survey methods, environmental data and statistical analyses. In a landscape-level monitoring approach, the spatial sampling units should preferably be oriented towards common international grids already used for pollinator or related monitoring programmes. Their extent should be chosen according to survey methods and the habitat ranges of the monitored pollinator organisms. Furthermore, sampling units should be placed in a design-based approach realised with random sampling. The required sample size can be calculated by power analyses based on the expected values and variances of pollinator species abundance and richness.

The case study on the monitoring of cavity-nesting wild bees in the agricultural landscapes of Germany has illustrated how the presented six-step guideline can be applied. The spatial sampling design allows landscape-scale analyses of wild bee assemblages and supports future synergies with related European monitoring initiatives. The specific future implementation of the long-term monitoring of wild bees in the agricultural landscapes of Germany depends on funding for the scientific support and data analysis.

## Figures and Tables

**Figure 1 insects-15-00229-f001:**
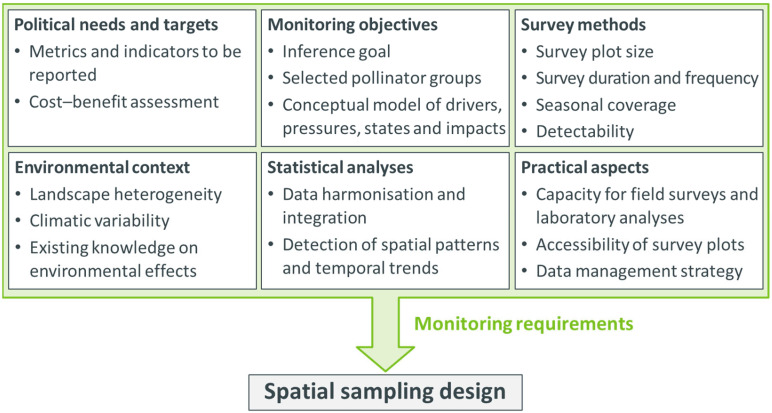
Requirements for the spatial sampling design of a large-scale pollinator surveillance monitoring programme. All listed requirements are essential for decisions in the spatial sampling design and need to be clarified in advance.

**Figure 2 insects-15-00229-f002:**
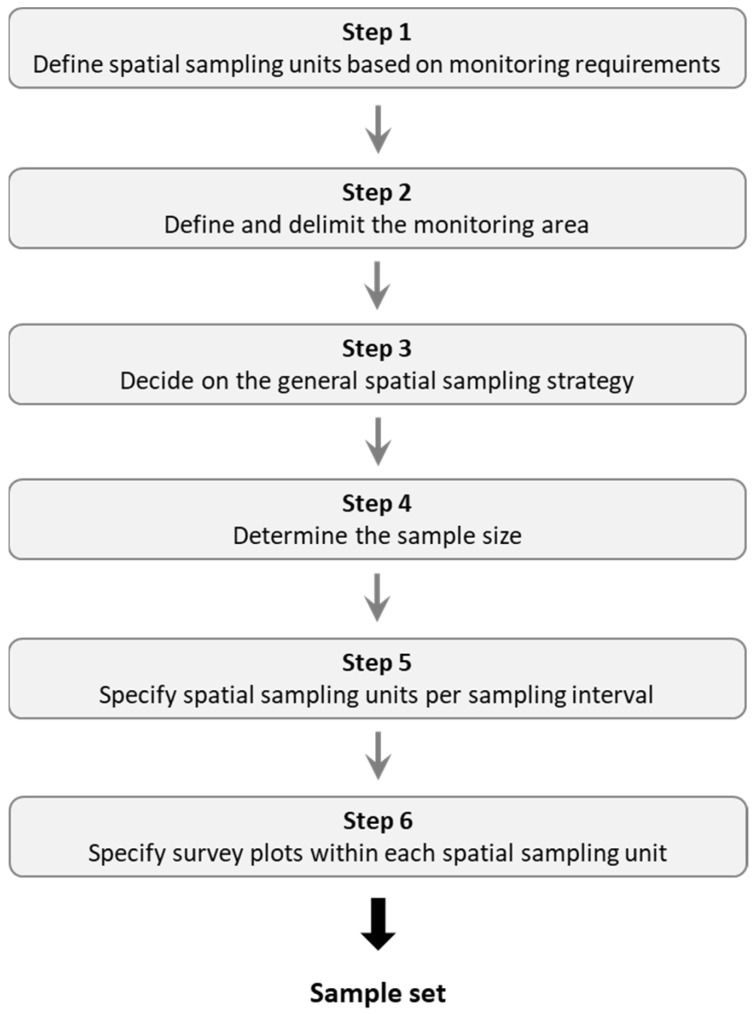
Six steps to achieve a representative spatial sample set for large-scale long-term pollinator monitoring.

**Figure 3 insects-15-00229-f003:**
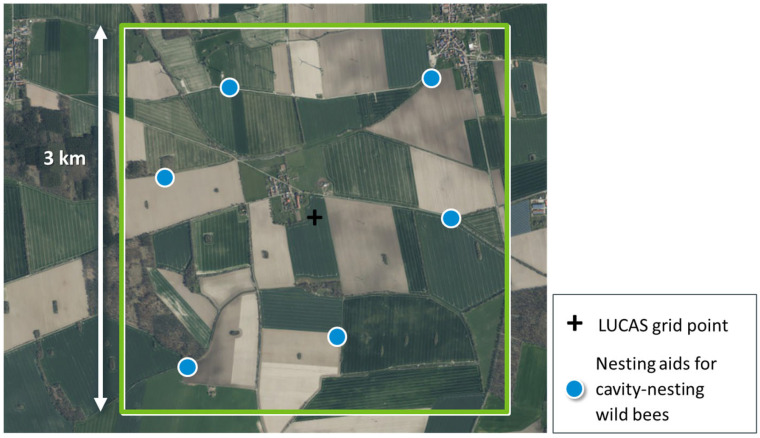
Spatial sampling unit defined as a landscape quadrat (3 × 3 km^2^, green frame) oriented towards the LUCAS grid (see Step 1). Locations of nesting aids for cavity-nesting wild bees are defined in Step 6 (Section 3.6.). Base map: DOP20 (BKG).

**Figure 4 insects-15-00229-f004:**
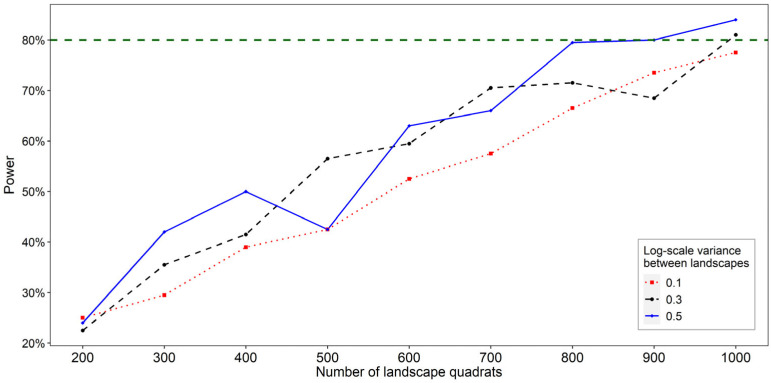
Results of the power analysis for a 3% decline in cavity-nesting wild bee species in nesting aids. Results are per sampling interval as dependent on the variance of species richness between landscape quadrats based on 200 simulation runs of generalised linear mixed models. The mean expected value was set to 15 species per landscape quadrat.

**Figure 5 insects-15-00229-f005:**
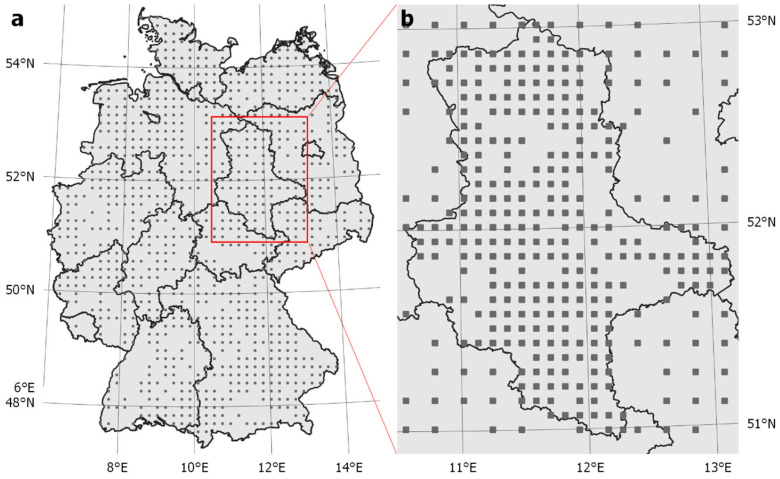
Locations of landscape quadrats: (**a**) sample of 950 landscape quadrats for monitoring cavity-nesting wild bees in agricultural landscapes of Germany based on a grid width of 16 km, (**b**) extended sample based on a grid width of 8 km for federal state-specific requirements using Saxony-Anhalt as an example. Base map: GeoBasis-DE (BKG).

**Table 1 insects-15-00229-t001:** Terminology used in the spatial sampling design of a landscape-level pollinator surveillance monitoring programme.

Term	Explanation	Example (Referring to Case Study Presented in Section 3)
Monitoring area	Spatial frame that defines criteria for sampling units to be considered suitable for inclusion in the sample set	Agricultural landscape of Germany, defined as at least 30% agricultural area per landscape quadrat
Sample set	All sampling units where the monitoring will be realised	950 sites with survey locations (Section 3.5.) and survey times
Sampling unit	Landscape segment with any number of pollinator surveys observed at one or more survey plots inside the sampling unit over a specific time	Landscape quadrat oriented towards the LUCAS grid with nesting aids observed over one season (Section 3.1.)
Survey	Collection of species data following a predefined method at a specific survey plot and time	Collection of data on cavity-nesting wild bees by taking photos of a pair of nesting aids at a specific location and time
Survey plot	Location of a survey with a predefined spatial extent	Location of a pair of nesting aids
Sampling strategy	Random or model-based method to select sampling units for the sample set	Systematic random sampling
Sample size	Number of sampling units included in the sample set	950
Sampling interval	Time between the beginning of two consecutive sampling periods, i.e., the temporal resolution of the monitoring	2 years

## Data Availability

Data supporting the findings of this study are available in the Appendix A of this article.

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
