# Peer review of "Six Steps towards a Spatial Design for Large-Scale Pollinator Surveillance Monitoring"

_insects, 2024, doi:10.3390/insects15040229_

Round 1

Reviewer 1 Report

Comments and Suggestions for Authors

The authors explain a conceptual framework for the sampling design of large-scale surveillance monitoring of pollinators and present a case study in Germany.

This study will be interesting for readers of Insects.

However, if the authors aim to develop a conceptual framework for facilitating the sampling design of large-scale surveillance monitoring of pollinators, more discussion is needed in the manuscript.

Major comments

Scale

The authors did not write about the relationship between the spatial scale of sampling units and pollinator characteristics.

Pollinator characteristics important in determining the scale, such as dispersal range, should be discussed in the manuscript.

Unexpected data loss

In large-scale monitoring, as the number of sampling sites increases, there is a possibility that data collection failures due to accidents will increase. The authors should also discuss strategies to mitigate the negative effects of unexpected data loss.

environmental DNA (eDNA) noise

DNA metabarcoding results include noise in the processes from extraction to PCR, library preparation processes, and index-hopping of artificial base sequences to identify derived samples due to reactions within the sequencer. There is a risk that a small amount of high-concentration DNA in one sample may be mixed into another in a laboratory. The methods of denoising have been proposed in several previous studies.

The authors should discuss strategies to mitigate the negative effects of eDNA noise.

Minor comments

Line 157

The authors should explain more details about LUCAS grid.

Author Response

***We have added our responses to the reviewer's comments below the respective comments. Line numbers refer to the revised version of the manuscript. Please see the attached PDF file for a formatted version of our response to reviewers.

The authors explain a conceptual framework for the sampling design of large-scale surveillance monitoring of pollinators and present a case study in Germany.
This study will be interesting for readers of Insects.
However, if the authors aim to develop a conceptual framework for facilitating the sampling design of large-scale surveillance monitoring of pollinators, more discussion is needed in the manuscript.
***Many thanks for the feedback and helpful comments! From the major comments, we understand that the focus of our manuscript was not clear enough. Our framework is a guideline for spatial sampling in large-scale pollinator surveillance monitoring; thus, we have specified our focus by replacing “sampling design” with “spatial sampling design” where it seemed reasonable. Moreover, we have added more discussion on applicability across pollinator groups and on the relationship between the spatial scale of sampling units and foraging range (see our response to the next comment).

Major comments

Scale
The authors did not write about the relationship between the spatial scale of sampling units and pollinator characteristics.
Pollinator characteristics important in determining the scale, such as dispersal range, should be discussed in the manuscript.
*** We have added a paragraph in the Discussion including more details on the relationship between the spatial scale of sampling units and foraging range: “Pollinators include a large diversity of insect taxa. This diversity is reflected by a large variety of habitat requirements in terms of nesting or oviposition sites, food preferences, and mating sites [80–82]. Different pollinator groups have been identified as suitable indicator taxa for large-scale monitoring schemes, for example solitary bees, bumblebees, butterflies, and hoverflies [83–85]. Our framework can be applied to all those taxa, with different decisions in the spatial sampling design process. For example, the size of the sampling units determines the spatial scale at which landscape effects can be evaluated. Therefore, sampling units should be large enough to cover the foraging range of species of interest. Landscape effects on pollinators have nevertheless been demonstrated at various spatial scales independent of their foraging range [33,86,87].” (Lines 414-423)

Unexpected data loss
In large-scale monitoring, as the number of sampling sites increases, there is a possibility that data collection failures due to accidents will increase. The authors should also discuss strategies to mitigate the negative effects of unexpected data loss.
***We agree that data loss is an important issue that has to be considered also during the design of a pollinator surveillance monitoring. However, our framework focusses on the spatial sampling design; thus, it seems not appropriate to focus on data loss within the six steps. Therefore, in the revised version, we have included “data management strategy” as bullet point in Figure 1 (under “practical aspects”) and added a sentence on this: “A strategy for data storage and management should be conceived (e.g., as a database management system) to allow efficient workflows and prevent data loss.” (Lines 93-95)

environmental DNA (eDNA) noise
DNA metabarcoding results include noise in the processes from extraction to PCR, library preparation processes, and index-hopping of artificial base sequences to identify derived samples due to reactions within the sequencer. There is a risk that a small amount of high-concentration DNA in one sample may be mixed into another in a laboratory. The methods of denoising have been proposed in several previous studies.
The authors should discuss strategies to mitigate the negative effects of eDNA noise.
***We agree that eDNA noise is important to be considered in the design of survey methods. However, our framework focusses on the spatial sampling design, and the survey methods belong to the decisions that need to be made prior to the spatial sampling design (see Figure 1 and Lines 79-81). Therefore, we think it would be confusing to add details on limitations of specific survey methods.

Minor comments

Line 157
The authors should explain more details about LUCAS grid.
***We have added a few more details: “Across Europe, the regular grid of the Land Use and Coverage Area frame Survey (LUCAS grid) is a useful basis for the definition of landscape quadrats as sampling units. The current LUCAS grid consists of 1,033,759 sample points, covering all member states of the European Union in a grid width of 2 km [42].” (Lines 158-161)

Reviewer 2 Report

Comments and Suggestions for Authors

The manuscript presents six steps in the development of a pollinator monitoring programme and therefore contributes to highly relevant topics considering pollinator decline and the development of effective insect monitoring. This is an important contribution especially considering the current development of an EU pollinator monitoring programme. The presented monitoring programme is scientifically robust and considers the substantial costs associated with such a program by employing a cost-effect method for observing wild bees using citizen science.

While the manuscript in many ways is convincing, the language is often convoluted and the wording confusing. The clarity of the manuscript would benefit a lot from a simpler writing style. This includes sentences such as: “Additionally, multiple dimensions of requirements need to be considered, all representing decisions that need to be made prior to the sampling design” (line 77-78); “we suggest the presented six-step procedure as a guideline to facilitate the sampling design as one of the essential parts in the development of large-scale pollinator” (446-448). Consider rewriting in a simpler language throughout the manuscript.

My only real concern is that the manuscript lacks a clearly defined focus. This should be easy to fix with some clearer language and definitions. The authors have taken a methodological angle, presenting a six step approach for developing monitoring programs, using the German monitoring programme as a case study. This is an important perspective that could provide valuable insight in the development of other monitoring schemes. As the authors write, when setting up large-scale monitoring there are many important decisions that must be made. Here, six steps in sampling design is presented/discussed. However, I don’t see a clear logic in 1) the steps the authors have chosen to focus on or 2) in the order of the steps as presented.

1) In the introduction (lines 77-97) the authors mention that many decisions have to be made prior to the development of the “sampling design”. This includes decisions that are not clearly distinguishable from what would traditionally include in a sampling design such as “taxonomic resolution, plot size, survey duration and frequency, seasonal coverage, and species detectability”. They then go on to explain (102-107) that “once these requirements have been clarified” the “sampling design itself” includes, as I understand it, spatial and some temporal aspects of the sampling design: for example, the number, size and spatial distribution of sampling plots and sampling intervals. Thus, there is not an intuitive distinction between parts of the design that is considered relevant in the six step process and those that are not. The definition of sampling design in Table 1 is also unclear. A sampling design is not “the procedure to define” a plan for sampling but the plan itself. It is also not clear what they mean by “organisational components” in terms of sampling design. Taken all together, this leaves a lot of room for confusion about what the authors have chosen to focus on in their manuscript and why. This needs to be clarified.

2) There is no clear logic in presenting issues regarding sampling units in step 1, step 5 and step 6. In the case study, for example, the random placement of nests/sampling plots is then described across multiple steps including 1, 3 and 6. When reading it, questions about why a random approach arises at step 1 but is not explained until step 6. That is a writing structure problem.

Below are some minor comments that can hopefully help improve the clarity of the manuscript.

Title: This is unusual wording and makes it quite unclear what the focus of the paper is.

Line 32-33: there is some confusion with the wording. “As a case study, we apply this framework to the recently developed and implemented wild bee monitoring in agricultural landscapes of Germany.” The steps presented is for developing the programme, so how can the method be applied to a program that is already developed and implemented?

65-66: “However, in particular for non-statisticians, further explanation is needed to develop a scientifically valid sampling design”. Unclear what you are trying to convey with this sentence.

72-74: this is a very long sentence that is hard to follow.

 91-94: this is regarding sufficient sample size that is also dealt with in step 4. Why is this mentioned in this paragraph which is about the decisions not included in the six steps?

Line 105: unclear what needs you are referring to.

Line 126: “commonly used terminology” This is too vague. In what context? By who and for what?

Line 132: here you refer to the overall design of the monitoring programme. What do you consider included in this design? A bit confusing since sampling design is also a term used often without much clarity.

Line 164-165: again the distinction between these steps and the rest of the development of the monitoring program is unclear. Figure 1

164-165: What decisions are made in this step? The political/monitoring objective are requirements determined the monitoring area (Fig 1). Why is this step included? Is it then not just one of the decisions that are made prior to the development of the study design? The very brief description of this step also seems to indicate that there is not many additional decisions to be made here.

166-168: it is unclear what is meant by “clear criteria to preselect sampling units”. This step is about the monitoring area.

170-171: “population of sampling units” and “a sample set out of that population”. It is confusing to use population both in terms of sample units and species. I suggest rewording.

183-185: It is unclear what is meant by “population parameters” in this sentence.

213: Is it true that sampling intervals is often one year in monitoring programs? There are no references to support this statement. I would think that would be overly expensive and probably not the time scale where substantial changes would happen due to for example land use change.

218-228: The description of this step appears superficial. How are sampling units “more or less randomly allocated”? If not completely random, what are the criteria?

312: When determining the sample size the required number of sample units is determined. But what about the number of traps within each sample unit. The number of traps will be important for accurately representing the local community and could vary with for example local richness. Are there not important decisions to be made about this and why are these decisions not considered equally important in the sampling design?

366: it is not explained why this step is taken in the random selection of survey plots.

433-445: How does this program align with EUPoMS? Is it complementary?

442: “meanwhile” is an odd word to use in this context. Consider rephrasing.

453: “Furthermore, monitoring data can lead to a better understanding of causes of pollinator declines as well as the role of land-use and environmental change.” This sentence is incomplete. The role of land use … change in this decline?

471: “The sampling design matches the landscape approach intended for the analysis of wild bee assemblages…” This is a confusing sentence. Which landscape approach?

Comments on the Quality of English Language

As mentioned in my general comments, a simpler language would help the authors get their points across more clearly. I suggest rewriting the manuscript with that in mind.

Author Response

***We have added our responses to the reviewer's comments below the respective comments. Line numbers refer to the revised version of the manuscript. Please see the attached PDF file for a formatted version of our response to reviewers.

The manuscript presents six steps in the development of a pollinator monitoring programme and therefore contributes to highly relevant topics considering pollinator decline and the development of effective insect monitoring. This is an important contribution especially considering the current development of an EU pollinator monitoring programme. The presented monitoring programme is scientifically robust and considers the substantial costs associated with such a program by employing a cost-effect method for observing wild bees using citizen science.
***Many thanks for the positive feedback and helpful comments!

While the manuscript in many ways is convincing, the language is often convoluted and the wording confusing. The clarity of the manuscript would benefit a lot from a simpler writing style. This includes sentences such as: “Additionally, multiple dimensions of requirements need to be considered, all representing decisions that need to be made prior to the sampling design” (line 77-78); “we suggest the presented six-step procedure as a guideline to facilitate the sampling design as one of the essential parts in the development of large-scale pollinator” (446-448). Consider rewriting in a simpler language throughout the manuscript.
*** We thank the reviewer to address the language. For the revised version of the manuscript, we have had the language checked by a native speaker and changed the text accordingly. We think that our points can be understood more clearly now.

My only real concern is that the manuscript lacks a clearly defined focus. This should be easy to fix with some clearer language and definitions. The authors have taken a methodological angle, presenting a six step approach for developing monitoring programs, using the German monitoring programme as a case study. This is an important perspective that could provide valuable insight in the development of other monitoring schemes. As the authors write, when setting up large-scale monitoring there are many important decisions that must be made. Here, six steps in sampling design is presented/discussed. However, I don’t see a clear logic in 1) the steps the authors have chosen to focus on or 2) in the order of the steps as presented.
***Please see our responses to the following two comments.

1) In the introduction (lines 77-97) the authors mention that many decisions have to be made prior to the development of the “sampling design”. This includes decisions that are not clearly distinguishable from what would traditionally include in a sampling design such as “taxonomic resolution, plot size, survey duration and frequency, seasonal coverage, and species detectability”. They then go on to explain (102-107) that “once these requirements have been clarified” the “sampling design itself” includes, as I understand it, spatial and some temporal aspects of the sampling design: for example, the number, size and spatial distribution of sampling plots and sampling intervals. Thus, there is not an intuitive distinction between parts of the design that is considered relevant in the six step process and those that are not. The definition of sampling design in Table 1 is also unclear. A sampling design is not “the procedure to define” a plan for sampling but the plan itself. It is also not clear what they mean by “organisational components” in terms of sampling design. Taken all together, this leaves a lot of room for confusion about what the authors have chosen to focus on in their manuscript and why. This needs to be clarified.
***We understand that the terminology may have caused some confusion. In Table 1, we tried to summarize the terms used in our framework, but we agree that the term “sampling design” mostly includes much more than the spatial design. Our framework is a guideline for spatial sampling in large-scale pollinator surveillance monitoring; thus, we have worked on the terminology for the revised version. We deleted the confusing definition of “sampling design” in Table 1. Moreover, we have specified our focus by replacing “sampling design” with “spatial sampling design” where it seemed reasonable.

2) There is no clear logic in presenting issues regarding sampling units in step 1, step 5 and step 6. In the case study, for example, the random placement of nests/sampling plots is then described across multiple steps including 1, 3 and 6. When reading it, questions about why a random approach arises at step 1 but is not explained until step 6. That is a writing structure problem.
***We understand that the general procedure might be a bit hard to follow. However, from the presentation of the framework in Section 2, we think that the steps are clear. Step 1 is just to define how the spatial sampling units look like (e.g., size and orientation of landscape segments). Of course, this depends on the target pollinator group and the spatial scale at which these pollinators should be analysed. The main task in the spatial sampling design is to get the sample set out of all possible sampling units, and this is achieved in Step 5. The sample set can be specified based on the preselection from Step 2, sampling strategy from Step 3, and the calculated sample size from Step 4. In Step 6, the survey plots within each sampling unit in the sample set are selected, i.e., the locations of the pollinator surveys within the selected landscape quadrats. We present this procedure in Section 2.
Regarding the case study, we agree that there might be some confusion due to redundancies between our descriptions of the steps. In the revised version, we have shifted the details on survey plots from the description in Step 1 (Section 3.1.) to Step 6 (Section 3.6.), and we have shifted details on the distances between landscape quadrats from Step 3 (Section 3.3.) to Step 5 (Section 3.5.). Hopefully, this makes also the general procedure in Section 2 better understandable.

Below are some minor comments that can hopefully help improve the clarity of the manuscript.

Title: This is unusual wording and makes it quite unclear what the focus of the paper is.
***We have modified the title to clarify that the presented six-step approach focusses on defining the spatial aspects of the sampling design.

Line 32-33: there is some confusion with the wording. “As a case study, we apply this framework to the recently developed and implemented wild bee monitoring in agricultural landscapes of Germany.” The steps presented is for developing the programme, so how can the method be applied to a program that is already developed and implemented?
***We have modified the sentence to avoid such misunderstandings: “As a case study, we apply this framework to the “Wild bee monitoring in agricultural landscapes of Germany”.” (Line 32-33)

65-66: “However, in particular for non-statisticians, further explanation is needed to develop a scientifically valid sampling design”. Unclear what you are trying to convey with this sentence.
***We have modified the sentence: “However, for non-statisticians in particular, further explanation is needed to develop a scientifically robust spatial sampling design.” (Lines 63-64)

72-74: this is a very long sentence that is hard to follow.
***We have modified the sentence: “Several questions need to be addressed in the sampling design [22], i.e.: where, when and how often to sample? What and how to sample? By whom and at which cost?” (Lines 69-70)

 91-94: this is regarding sufficient sample size that is also dealt with in step 4. Why is this mentioned in this paragraph which is about the decisions not included in the six steps?
***We agree that this could have caused misunderstandings. We modified this part: “Fifth, the intended statistical analysis of pollinator trends requires preliminary considerations on the statistical power for trend detection as a basis for determining the sample size. Moreover, to define spatial and temporal scales of the sampling design, the planned level of harmonisation of pollinator and environmental data needs to be clarified in advance.” (Lines 87-91)

Line 105: unclear what needs you are referring to.
***We have modified this sentence: “Inspired by previous studies [19–21], we present a six-step approach to provide taxonomists and conservationists with a guideline for the development of a representative spatial sample set corresponding to their own set of requirements (according to Figure 1).” (Lines 102-105)

Line 126: “commonly used terminology” This is too vague. In what context? By who and for what?
***We agree and have modified the sentence. The terminology is used in the context of our study: “Following the terminology in spatial sampling introduced in Table 1, ...” (Line 125)

Line 132: here you refer to the overall design of the monitoring programme. What do you consider included in this design? A bit confusing since sampling design is also a term used often without much clarity.
***We have modified this sentence: “This six-step procedure for the spatial sampling design should be considered as part of the overall development of the monitoring programme, including all steps according to Reynolds et al. [21] and the clarification of monitoring requirements (Figure 1).” (Lines 130-133).

Line 164-165: again the distinction between these steps and the rest of the development of the monitoring program is unclear. Figure 1
***To allow a better distinction between the requirements from Figure 1 and the six-step procedure, we have clarified the focus of our study (“spatial sampling design”). The predefined monitoring objectives and political context (according to Figure 1) are important for the decisions that need to be made here in Step 2. To better understand the decisions made in Step 2, we have reformulated this paragraph (see our response to the next comment).

164-165: What decisions are made in this step? The political/monitoring objective are requirements determined the monitoring area (Fig 1). Why is this step included? Is it then not just one of the decisions that are made prior to the development of the study design? The very brief description of this step also seems to indicate that there is not many additional decisions to be made here.
***We have reformulated this paragraph to clarify the decisions made in Step 2:
“The area for pollinator monitoring needs to be defined according to the predefined monitoring objectives and political context. For example, this could be the total area of a specific country or physiographic region, or agricultural landscapes within a specific region. Furthermore, considerations on survey methods, environmental context and practical aspects may require a specific preselection of the sampling units from Step 1. It is crucial to define clear preselection criteria on the location and landscape composition of sampling units. Based on such criteria, the result of Step 2 is a subset of the sampling units determined in Step 1 (i.e., all sampling units that fulfil the criteria), which is then further considered in Steps 3-6.” (Lines 166-174)

166-168: it is unclear what is meant by “clear criteria to preselect sampling units”. This step is about the monitoring area.
***That’s exactly what those criteria should be about. The criteria should be defined to select sampling units corresponding to the monitoring area. We hope that this is clearer in the revised version (see our response to the previous comment).

170-171: “population of sampling units” and “a sample set out of that population”. It is confusing to use population both in terms of sample units and species. I suggest rewording.
***”Population” is the common statistical term that describes the result of Steps 1 and 2 (and there is no other common term fitting in this context). However, we understand that it could lead to misunderstandings, so we have reformulated this sentence: “After the sampling units have been defined in Steps 1 and 2, a general strategy is needed to select a sample set.” (Lines 176-177)

183-185: It is unclear what is meant by “population parameters” in this sentence.
***We have modified this sentence: “… that explains the distribution of the population parameters such as diversity or abundance based on environmental data.” (Lines 190-191)

213: Is it true that sampling intervals is often one year in monitoring programs? There are no references to support this statement. I would think that would be overly expensive and probably not the time scale where substantial changes would happen due to for example land use change.
***We agree and have modified this sentence: “Sampling intervals could for example be defined as one year to optimally cover interannual variation [62].” (Lines 219-220)

218-228: The description of this step appears superficial. How are sampling units “more or less randomly allocated”? If not completely random, what are the criteria?
***We have specified the description: “The exact locations of pollinator surveys need to be specified according to the survey methods, so that acquired pollinator data are representative of the whole landscape. For example, to receive a good representation of the present pollinators, a single transect with its linear shape should be placed crossing as many habitat types as possible according to their presence in the sampling unit. As another example, multiple point locations are needed in the case of any kind of trap. Their number per sampling unit should be high enough to represent the pollinator community at the landscape level, i.e., the scale of the sampling units. Across every sampling unit, traps should be located either randomly or in a model-based approach based on landscape features, according to the predefined monitoring objectives. Depending on the target pollinators, a minimum distance between survey plots needs to be kept to count as independent.” (Lines 227-237)

312: When determining the sample size the required number of sample units is determined. But what about the number of traps within each sample unit. The number of traps will be important for accurately representing the local community and could vary with for example local richness. Are there not important decisions to be made about this and why are these decisions not considered equally important in the sampling design?
***The conceptual framework refers to large-scale surveillance monitoring programmes with sampling units at the landscape scale. This means that pollinator data are always collected at the landscape level, and the sample size refers to the number of landscape-scale sampling units. Depending on the size of the landscape segments and on the survey method, it could be necessary to install more than one survey plot within every landscape segment. For the example of traps, we agree that it is important to consider how many plots are needed, but this is not a question of the spatial sampling design, but rather a question of specific survey methods. This should be clear from the beginning, before the spatial sampling units are defined. To underline the role of the number of survey plots for the example of traps, we have added a sentence in Section 2.6.: “Their number per sampling unit should be high enough to represent the pollinator community at the landscape level, i.e., the scale of the sampling units.” (Lines 232-234)

366: it is not explained why this step is taken in the random selection of survey plots.
***We have added a sentence as explanation: “Step (E) ensures that all survey plots are arranged with distances > 500 m such that cavity-nesting wild bees are surveyed independently.” (Lines 368-369)

433-445: How does this program align with EUPoMS? Is it complementary?
***At the moment, EUPoMS is still in a conceptual phase. With our focus on cavity-nesting bees in the agricultural landscape of Germany, we only cover one group of pollinators, and there are more to be covered by EUPoMS. Thus, we are currently not part of a European approach, but in close dialogue with the EUPoMS colleagues. The sampling design of EUPoMS is planned based on the LUCAS grid (see Potts et al. 2021 [10]). Our spatial sampling framework can thus serve as an example, how a European approach such as EUPoMS may be scaled to (sub)programmes of at the national levels.

442: “meanwhile” is an odd word to use in this context. Consider rephrasing.
***We have modified this sentence: “For example, field data on pollinators could be connected to the growing amount of easily available data from remote sensing and other earth observation data [95–97].” (Lines 453-455)

453: “Furthermore, monitoring data can lead to a better understanding of causes of pollinator declines as well as the role of land-use and environmental change.” This sentence is incomplete. The role of land use … change in this decline?
***We have modified this sentence: “Furthermore, monitoring data can lead to a better understanding of causes of pollinator declines and their relationship to land-use change or environmental change.” (Lines 463-465)

471: “The sampling design matches the landscape approach intended for the analysis of wild bee assemblages…” This is a confusing sentence. Which landscape approach?
***We have modified this sentence: “The spatial sampling design allows landscape-scale analyses of wild bee assemblages and supports future synergies with related European monitoring initiatives.” (Lines 482-483)

Comments on the Quality of English Language
As mentioned in my general comments, a simpler language would help the authors get their points across more clearly. I suggest rewriting the manuscript with that in mind.
***We have had the language checked by a native speaker and changed the text accordingly. We think that our points can be understood more clearly now.

Round 2

Reviewer 1 Report

Comments and Suggestions for Authors

The title of the Supplementary Materials is different from the title of the main text.

Please check the revised parts of the main text again and ensure consistency.

Author Response

Many thanks! We have corrected the title in the Supplementary Materials and checked the revised parts of the main text again.